# Towards Improved Bioavailability of Cereal Inositol Phosphates, *Myo*-Inositol and Phenolic Acids

**DOI:** 10.3390/molecules30030652

**Published:** 2025-02-01

**Authors:** Krzysztof Żyła, Aleksandra Duda

**Affiliations:** 1Department of Biotechnology and General Technology of Foods, Faculty of Food Technology, University of Agriculture in Krakow, ul. Balicka 122, 30-149 Krakow, Poland; 2Department of Fermentation Technology and Microbiology, Faculty of Food Technology, University of Agriculture in Krakow, ul. Balicka 122, 30-149 Krakow, Poland

**Keywords:** cereals, phytic acid, ferulic acid, phytase, inositol phosphates, feruloyl esterase, tight junctions, GLP-1, P-glycoprotein inhibition, enzymatically generated inositol (EGI)

## Abstract

Cereals are among the foods rich in *myo*-inositol hexakisphosphate (phytic acid, IP6), lower *myo*-inositol phosphates (IPx), a wide range of phenolic compounds, as well as vitamins, minerals, oligosaccharides, phytosterols and para-aminobenzoic acid, and are attributed with multiple bioactivities, particularly associated with the prevention of metabolic syndrome and colon cancer. The bran fraction of wheat, maize, brown rice and other cereals contains high levels of phytate, free and total phenolics, and endogenous enzymes such as amylases, phytase, xylanase, β-glucanase and feruloyl esterase, whose activities can be increased by germination. The preliminary steps of digestion begin in the oral cavity where substrates for the action of endogenous cereal and salivary enzymes start to be released from the food matrix. IP6 released from phytate complexes with arabinoxylans, starch and protein bodies would eventually enhance the absorption of nutrients, including phenolics, by regulating tight junctions and, together with ferulic acid (FA), would maintain cell barrier integrity and epithelial antibacterial immunity. In addition, both IP6 and FA exert potent and complementary antioxidant effects, while FA together with IPx generated through advanced hydrolysis of IP6 by endogenous and microbial phytases may affect digestive enzyme activity and incretin secretion, resulting in modulated insulin and glucagon release and prevention of various diabetic complications. Contrary to widespread negative attitudes towards phytate, in this review, we present the strategy of selecting cereals with high phytate and phenolic content, as well as high endogenous phytase, feruloyl esterase and endoxylanase activities, to produce value-added health-promoting foods. The advanced hydrolysis of phytate and phenolic compounds by cereal and/or microbial enzymes would generate substantial amounts of “enzymatically generated inositol” (EGI), including IP6, IPx and *myo*-inositol, the compounds that, together with free FA, provide enhanced bioavailability of cereal nutrients through multiple synergistic effects not previously realised.

## 1. Introduction

A scoping review of the Nordic Dietary Recommendations for Cereals and Cereal Products [1] provides summaries of the associations between whole grain consumption and important health outcomes, with convincing evidence of reduced risk of coronary heart disease, colorectal cancer, premature mortality and incidence of type 2 diabetes. The randomised trials reviewed in this article also showed moderate effects on changes in body weight, total cholesterol and systolic blood pressure. Components of whole grains that have been reported to be bioactive include minerals, B vitamins, vitamin E, iron, magnesium, selenium and fibre. Fardet [2], attempting to answer the question “What is beyond fibre?”, published an extensive list of wheat phytochemicals and discussed numerous possible biological mechanisms in which they are involved, as well as their effects on human health. The bran and germ fractions of cereals are sources of vitamins, minerals, carotenoids, trace elements, polyphenols, phytate, as well as oligosaccharides, phytosterols, para-aminobenzoic acid and α-linolenic acid, among others. Most studies to date have focused on determining the physiological effects of cereal compounds separately, overlooking potential synergies between different bioactive components and interactions of these substances with components of the cereal food matrix. In addition, different factors have been identified as critical in determining bioaccessibility in different segments of the intestine and, consequently, their bioavailability and bioactivity in tissues and organs. The complexity of food systems highlights the difficulties in assessing the potential health-promoting activity of an isolated tissue component and the need to contrast observed effects with those characteristic of the cereal as a whole.

Among foods, cereals, legumes, oilseeds and nuts have the highest levels of phytic acid/phytate [3]. The bran fraction of wheat and rice is particularly high in phytate, and, along with sesame, flaxseed, fava beans and almonds are usually included in a list of the richest natural sources of phytate [4,5]. Further, cereals, whole grains, nuts, almonds and legumes are among the foods rich in phenolic compounds in addition to phytate. Conversely, fruits, vegetables, cocoa, green tea, spices and microalgae are foods that are high in polyphenols but low in phytate. The diverse functional properties of phenolic compounds in cereals, especially the antioxidant activity of phenolic acids, flavonoids and lignans, have only recently been documented and appreciated [6,7]. It should be emphasised that, similar to phytate, the bran fraction of wheat, brown rice, maize and other cereals contains the majority of both free and total phenolics. Extracts of wheat bran and barley hulls with different polarities were rich in saponins, phenolics and phytate, with high antioxidant activities as determined by DPPH and ABTS assays [8]. In a discussion, the authors related the antioxidant activity of the extracts to the concentrations of phenolic compounds and flavonoids, but did not consider phytate as an antioxidant. Similarly, other studies have suggested that the breeding of nutritious cereals should focus on varieties with high phenolic bioavailability, as a diet rich in cereal phenolics may modulate microbial composition, support gut homeostasis and health, and thus provide a basis for the production of new functional cereal-based foods [9]. Other studies have suggested that raw materials with high phenolic concentrations could be subjected to bioprocessing, such as fermentation or enzymatic treatment to improve the bioaccessibility and bioavailability of bread phenolics [10]. The effects of milling, heat treatment, extrusion and germination and fermentation on cereal polyphenols, whose antioxidant activity was considered to be the main bioactivity of whole grains, were also discussed by Tian et al. [6]. However, the bioactivities resulting from high levels of phytate and its hydrolysis products seem to be unappreciated and overlooked in research. Bread-making recipes with different yeast proofing times or based on sourdough mixed cultures were found to be key factors that could minimise phytic acid content in breads made from flours of different wheat species with different endogenous phytase activities [11]. High phytate content in whole grains remains a controversial issue [12,13]. Many reviews have contrasted the nutrient-chelating properties of phytate, which impair mineral bioaccessibility, with the diverse health-promoting activities of phytate isolated from rice or maize. Previous studies reviewed the anticancer properties of pure phytate, pointing to unique new options for the treatment of different types of cancer [14,15,16]. Also in 2019, a case report described a patient with stage IV metastatic melanoma who refused conventional therapy, but was treated with phytic acid and *myo*-inositol (800/220 mg)—the patient achieved a complete remission that lasted for 3 years [17]. This and much other evidence of the therapeutic efficacy of phytic acid alone or in combination with *myo*-inositol in cancer treatment have been reported, including the alleviation of side effects of chemotherapy [18].

Oral administration of phytic acid solution enhanced the absorption of blackcurrant anthocyanins in rats and humans [19], as well as isorhamnetin, quercetin and kaempferol from *Hippophae rhamnoides* L. in an in vitro Caco-2 cell experiment. These effects were confirmed in in vivo pharmacokinetic studies [20]. In curcumin-loaded nanoemulsions, phytic acid improved droplet flocculation and the formation of mixed micelles from which curcumin was solubilised and made more bioavailable, while inhibiting lipid digestion [21]. It has been hypothesised that phytic acid may be actively involved in tight junction regulation, and therefore play a role as an absorption enhancer [22]. It has also been reported that tight junctions are regulated by natural polyphenols [23]. This raises the intriguing possibility of interactions and synergies between cereal phenolics and phytate (*myo*-inositol phosphates), especially when whole-grain cereal products are consumed. In addition, these two bioactive components of cereals may be involved in maintaining the integrity of the cell barrier [24] and may modulate intestinal epithelial antibacterial immunity [25]. Both phenolics and phytate deserve more in-depth analysis for their possible role in influencing incretin secretion and, consequently, insulin and glucagon regulation [26]. To make the issue even more important, there have also been reports on the role of phenolics [27] and phytic acid [28] in inhibiting P-glycoprotein and, consequently, enhancing the bioactivity of other food components or drugs. In general, potential interactions between cereal phenolics and phytate degradation products do not seem to be recognised in the literature and related issues seem to be overlooked in research.

Here, we review and critically analyse data on the major cereal constituents’ phytate and phenolic compounds, with particular emphasis on factors that determine the bioaccessibility of phytate and phenolic compounds from different segments of the gastrointestinal tract. We have also focused on mechanisms of possible interactions between these phytochemicals that may maximise their beneficial effects by influencing their ultimate bioavailability and multiple bioactivities.

## 2. The Breakdown of Cereal Phytates and Polyphenols in Oral Cavity

Multiple mechanisms of interactions between salivary components and food components have only recently been recognised [29,30]. Apart from colloidal interactions, complexation, binding and release of aroma compounds, the enzymatic degradation of starch by salivary α-amylase deserves special attention, mainly due to its association with the development of metabolic syndrome [31]. Some studies have shown that the interactions of polyphenols with salivary amylase, and, in particular, the inhibition of the salivary enzyme, are limited to tannins, gallotanin and EGCG [32], while the other [33] reported that phenolic acids, both free and bound to quinic acid, are poor inhibitors of human salivary α-amylase. It should be noted that the results of the in vitro studies suggesting the ability of phenolic acids with a cinnamic acid backbone (caffeic acid, ferulic acid) to inhibit α-amylase were not performed in a simulated salivary environment [34,35]. In 1984, a study using a simple in vitro dialysis system, showed that the addition of tannic acid and phytic acid reduced starch digestibility by 13 and 60%, respectively, and that there were no additive effects when these compounds were combined [36]. Some years later, another study demonstrated the inhibition of human salivary α-amylase and starch digestibility by phosphate esters of *myo*-inositol [37]. The inhibition was linearly correlated with the degree of phosphorylation of the inositol ring and was also dependent on pH. Although the results of these in vitro studies must not be used to draw conclusions about in vivo bioactivities, recent comprehensive reviews [38,39] have reported a negative correlation between phytic acid content and the glycaemic index of various starchy crops. The type of inhibition exerted by phytic acid on human salivary α-amylase was found to be mixed non-competitive and pH-dependent [40]. On the other hand, it has been reported that the degradation of flavonols (kaempferol, quercetin, isorhamnetin and myricetin) starts in the presence of saliva and phenolic acids (4-hydroxybenzoic acid, protocatechuic acid, vanillic acid and gallic acid, respectively) are partially released [41].

It is important to note that during digestion in the mouth, food is subjected to mechanical breakdown by the teeth, including cutting, grinding, shearing and mixing with salivary salts, mucins and enzymes [42]. Although the role of salivary α-amylase and lingual lipase in starch hydrolysis and fatty acid release is usually considered, little is known about the enhanced functionality of endogenous food enzymes, which, like their substrates, are most likely partially released from complexes with other cell wall constituents. For example, disruption of the cereal food matrix in the mouth allows cereal phytase, β-glucanase and feruloyl esterase access to soluble phytate, β-glucan and ferulic acid, respectively [43], and salivary water increases water activity to a level sufficient for enzymatic action. Although the pH of saliva (6.8–7.0) does not appear to favour these enzymatic reactions, the initial stages of digesta acidification in the stomach may create a favourable environment (pH 4–5) for the onset of such reactions (Figure 1). This, together with the advanced physical disintegration of the tissues, makes the gastric digestion phase the most important for the catalytic action of these enzymes.

In a study comparing starch digestion in individuals with high and low endogenous salivary amylase activity, significantly higher plasma insulin concentrations and significantly lower postprandial blood glucose concentrations were observed in healthy, non-obese individuals with high salivary amylase activity [44]. The authors hypothesised that incretins, such as glucagon-like peptide-1, may be released peripherally from lingual taste cells into the bloodstream and stimulate insulin release from the pancreas.

## 3. Digestion in the Stomach

Mechanical action and gastric juice components such as hydrochloric acid, pepsinogens, mucus, lipase, electrolytes and water are key contributors to food digestion in the stomach [45]. Although the low pH (1.3–3.0) of gastric juice contributes to pepsin activation and protein denaturation (Figure 2), the cereal foods exert a buffering capacity and raise the pH in the stomach to around 3 or even 5 (depending on the initial particle size of the food, the initial moisture content, the time of digestion and the region of the stomach) [46]. This limits proteolysis and can activate endogenous phytase, β-glucanase and feruloyl esterase. On the other hand, in the acidic state of the stomach, phytate is quite soluble, allowing phytases to exert their catalytic action [47]. Detailed data on phytate degradation products in different parts of the human gastrointestinal tract are lacking, but Schlemmer and co-workers studied the mechanisms in pigs, which may serve as a perfect model for human digestion [48]. When the diet was high in intrinsic plant phytases, about 37–66% of dietary phytate was degraded in the pig stomach and small intestine. Products of plant 6-phytases, mainly IP5(1,2,3,4,5) and IP3(1,2,3), dominated the stomach, but the liberation of phytate from the dietary matrix was not complete. Yu and co-workers [49] investigated the ability of different *myo*-inositol phosphate esters (IP6-IP1-2) to aggregate proteins, chelate Fe^3+^ ions and inhibit pepsin. They found that to alleviate pepsin inhibition by phytate, IP6 had to be dephosphorylated to IP1-2 and that the reactivity of IPx towards Fe^3+^ and the effects on protein aggregation were proportional to the degree of *myo*-inositol phosphorylation. These in vitro studies were carried out at pH 2.5 and must therefore be considered with caution, as all reported effects are certainly pH-dependent. Pepsin inhibition by phytate may be accompanied by an increase in pepsin and mucin secretion, which increases endogenous amino acid fluxes, at least in chickens [47].

Unlike phytate, the effects of phenolic compounds on pepsin activity and gastric digestion are not straightforward. In early in vitro studies, resveratrol, catechin, epigallocatechin-3-gallate and quercetin were found to enhance pepsin digestion of different food proteins by increasing V_max_ and k_3_ of the reaction [50]. However, the study investigating the interactions between naringenin and pepsin using spectroscopic analysis and docking simulation [51] found that naringenin, a citrus flavonoid also found in cereals [52], altered the secondary structure of the enzyme, binding to its active site and, consequently, reducing pepsin activity. Generally, phenolic compounds are thought to be poorly absorbed and, remaining in the gastrointestinal tract, exert inhibitory effects not only on digestive proteinases but also on amylases and lipases [53]. The authors concluded that while the inhibition of lipid and starch digestion may beneficially affect energy metabolism in certain dietary situations, deteriorations in protein digestion may compromise the bioaccessibility and bioavailability of essential amino acids.

From this point of view, phenolics must be considered as antinutrients, especially in plant-based diets. Notably, the processing technologies suggested by the authors to improve the antinutritional properties of polyphenols are exactly the same as those that mitigate the antinutritional effects of phytate. Hydrophilic cereal phytochemicals such as phenolic acids are thought to be available for absorption in the stomach, provided they are not tightly bound to fibre, which would compromise their absorption [54]. Furthermore, these compounds may also be subject to hydrolysis and deconjugation in the stomach [55]. Duda-Chodak and Tarko [56] reviewed the possible negative and side effects of polyphenols in the human diet and pointed out many negative consequences of the decrease in digestive enzyme activity, especially in elderly subjects. On the other hand, phenolic acids were claimed to have no effect on nutrient digestion. Li and co-workers [57] investigated the cellulase-assisted ultrasonic treatment of brown rice prior to germination to maximise the bioaccessibility of nutrients, including phenolics and phytate. Although these studies did not separate the effects of cellulase treatment from those of sonication, the combined effects of ultrasound-assisted cellulose treatment on brown rice grains were profound. Changes in grain surface structure resulted from increased water absorption during soaking, which contributed to improved germination rates, and increased activity of phytase, glutamate decarboxylase, succinate semialdehyde dehydrogenase, gamma-aminobutyric acid (GABA) transaminase, chalcone isomerase and phenylalanine ammonia lyase. In addition to reduced phytate levels, the grains had higher levels of GABA, flavonoids and phenolics. Furthermore, in in vitro digestions, phenolics showed higher bioaccessibility than their counterparts in grains not subjected to enzyme-assisted ultrasound treatment. Ultrasound-assisted enzymatic pretreatment of cereal grains prior to germination and sprouting as a method to increase the bioaccessibility and bioavailability of phenolics, among other nutrients, and to stimulate phytate dephosphorylation offers a novel approach in cereal processing.

While lipolysis studies in most in vitro digestion models have been limited to pancreatic lipase active in the duodenum [58], the combined effect of gastric lipase with pepsin on commercial infant formulae has only recently been investigated [59]. A significant effect (up to 20 wt%) of gastric lipase on the overall gastrointestinal digestion of high lipid emulsions was reported by [60]. Long-term administration of rosemary extracts rich in carnosic acid to female Zucker rats decreased body weight, serum triglycerides, cholesterol, insulin levels and gastric lipase activity [61]. The effects of phytic acid on lipid digestion and the bioaccessibility of curcumin from oil-in-water nanoemulsions in the simulated gastrointestinal tract were investigated [21]. The authors found that phytic acid improved droplet flocculation, mixed micellar formation, curcumin solubilisation and bioaccessibility, but inhibited lipid digestion. It is therefore reasonable to speculate that gastric lipase activity may be modulated by both phenolics and inositol phosphates. Finally, it should be emphasised that digestion in the stomach has a profound effect on later stages of digestion, particularly due to mechanical effects and pH changes.

## 4. Digestion in the Small Intestine

### 4.1. Regulation of Gut Hormones—Incretins (GIP, GLP-1) by Phenols and Inositol Phosphates and Inhibitory Effect of These Compounds on α-Amylase and α-Glucosidase Activities

Intestinal secretions in the duodenum (HCO_3_, bile and pancreatic enzymes) neutralise the acidic chyme from the stomach, producing a pH of 6.2 in the duodenum, 6.7 in the jejunum and 7.4 in the ileum, and carry out the digestion and absorption of starch, fat and protein with the active participation of intestinal bacteria [62]. Specific cellular receptors on the surface of the epithelium, as well as changing levels of carbon dioxide and oxygen in different segments of the intestine, have been found to control intestinal homeostasis. In particular, in the duodenum, starch-digesting enzymes such as pancreatic α-amylase, mucosal α-glucosidase subunits—maltase–glucoamylase plus sucrose–isomaltase generate glucose, which is absorbed in the jejunum and ileum (Figure 3). The released nutrients, while interacting with the mucosa, stimulate the secretion of gastrointestinal hormones such as GLP-1, cholecystokinin and PYY [63]. Intestinal proteolysis is enhanced in the duodenum mainly by pancreatic proteases and peptidases, as well as brush border exopeptidases, which generate small peptides and amino acids [64]. Food processing techniques, by modifying the microstructure of foods, also have a profound effect on both protein digestion and absorption [65]. In cereal digestion, a number of food-derived protease inhibitors, including some that also inhibit α-amylase, influence digestive physiology [66,67], while emulsified proteins are subject to an interplay between proteolysis and lipolysis [68]. In human bile, the molar ratio of two biosurfactants, bile salts and phospholipids, critically influenced lipid dispersion [69] and, together with pancreatic lipase levels, influenced triglyceride hydrolysis and free fatty acid uptake by enterocytes [70].

There is considerable current interest in the interactions between the gut and the brain and in the health of other tissues (gut–brain, gut–liver axes). At the small intestinal level, multiple mechanisms influence the bioaccessibility and bioavailability of cereal nutrients, with phytates and phenolic compounds appearing to be involved in many modulating processes. Multilevel hormonal control exerted in the gut mainly by incretin peptides, i.e., glucose-dependent insulinotropic polypeptide (GIP) and glucagon-like peptide-1 (GLP-1), through direct and indirect actions on islet β-cells, regulate insulin secretion, glucagon secretion, glucose concentrations, lipid metabolism, appetite, body weight and immune function [26,71]. Both peptides lower blood glucose, slow nutrient absorption and are inactivated by dipeptidyl peptidase-4 (DPP-4) [72]. Natural agonists of the GLP-1 receptor (GLP-1R) and DPP-4 inhibitors are used clinically and are also important in foods with pro-health functions [73,74].

A systemic review and meta-analysis revealed that whole grain intake did not modulate GLP-1 levels, but increased postprandial GIP concentrations from 60 to 180 min after ingestion, particularly in participants with diabetes and metabolic syndrome [75]. In another paper, no consistent effect of soluble cereal fibre (oat β-glucans, resistant starch) on GLP-1 levels was demonstrated in healthy subjects, despite an increase in the levels of SCFAs generated from fibre during colonic fermentation [76]. No effects of type 2 resistant starch and β-glucans on other gastrointestinal hormones were reported, except for decreased ghrelin and increased PYY, which may explain the weight loss effects of soluble cereal fibre fractions. In another study on oat β-glucans, increased transit time and GIP, decreased appetite score, plasma ghrelin, insulin and glucose were observed, with no effect on GLP-1, PYY and glucagon [77]. A more comprehensive review of the role of cereal bioactive compounds in the prevention and treatment of type 2 diabetes was provided by Yilmaz et al. [78], who considered proteins, phenolic compounds, phytate, carotenoids and tocols, in addition to soluble and non-soluble fibre.

Both phenolics and phytate have been shown to modulate glycaemic response by inhibiting α-amylase and α-glucosidase, but, in addition, cereal phenolics promoted β-cell function and insulin secretion, while phytic acid improved colonic health by binding to various proteins and peptides and forming binary and tertiary structures. The phytate–protein and phytate–mineral–protein complexes, on the one hand, and the interactions of phytic acid with the digestion of starch, on the other hand, led to a slowing down of the gastric emptying process. It should be emphasised that cereals (including barley, maize and wheat) have been reported to contain various proteins capable of inhibiting amylolytic and proteolytic enzymes, such as bifunctional α-amylase/trypsin inhibitors, α-amylase inhibitors, limit dextrinase inhibitors and serine protease inhibitors [79], and barley peptides were claimed to inhibit not only α-amylase and α-glucosidase, but also DPP4 [80]. In addition, oat and barley β-glucans and arabinoxylans, due to their gel-forming properties, increased intestinal viscosity and, consequently, prolonged gastric emptying time, thereby reducing nutrient absorption [80].

Polyphenols extracted from different food sources such as sweet potato, coffee beans, cocoa beans, fermented cocoa beans, roasted cocoa liquor, anthocyanin-rich grape seed extract, polyphenol-rich cranberry extract and aqueous cinnamon extract have been reported to stimulate GLP-1 secretion, increase its half-life by inhibiting dipeptidyl peptidase-4 (DPP4), and stimulate β-cells to secrete insulin [81]. Similarly, a polyphenolic fraction from whole millet (*Setaria italica*) grains extracted at 70 °C with 60% ethanol using 400 W ultrasound, containing ferulic acid, p-coumaric acid, 2-hydroxycinnamic acid and coniferyl aldehyde, promoted endogenous GLP-1 secretion and prevented weight gain in diet-induced obese mice [82]. More recently, ferulic acid was found to bind to the GLP-1 molecule in a 1:1 ratio by van der Waals and electrostatic forces, effectively preventing hydrolysis of GLP-1 by DPP4 [83]. In addition, a study of glucose metabolism in mice fed a high-fat diet demonstrated that the combination of ferulic acid with arabinoxylan or with β-glucans increased serum GLP-1 levels and regulated the gut microbiota by increasing the abundance of *Bifidobacterium* and *Faecalibaculum* bacteria [84]. *Lactiplantibacillus plantarum*—a lactic acid bacterium found in the gastrointestinal tract of fish, insects and mammals, including humans, in addition to secreting many enzymes—also stimulates the synthesis of GLP-1 [85] (Figure 4A,B).

Glucagon-like peptide-1 is known to activate adenylyl cyclase, which converts ATP to cAMP, which, in turn, activates protein kinase A (PKA) (Figure 4A). PKA in the β-cell triggers insulin secretion by stimulating phosphoinositol turnover via phospholipase C-mediated synthesis of inositol triphosphate (IP3) and release of intracellular Ca^2+^. Diacylglycerol generated by phospholipase C, together with Ca^2+^ ions, activates protein kinase C (PKC), the enzyme that phosphorylates the C-terminus of GLP-1 by additional kinase action [86]. Hormones that activate phospholipase C, such as gastrin-releasing peptide and acetylcholine, have also been proposed to increase the production of IP3 and, consequently, the synthesis of GLP-1 [87,88].

**Figure 4 molecules-30-00652-f004:**
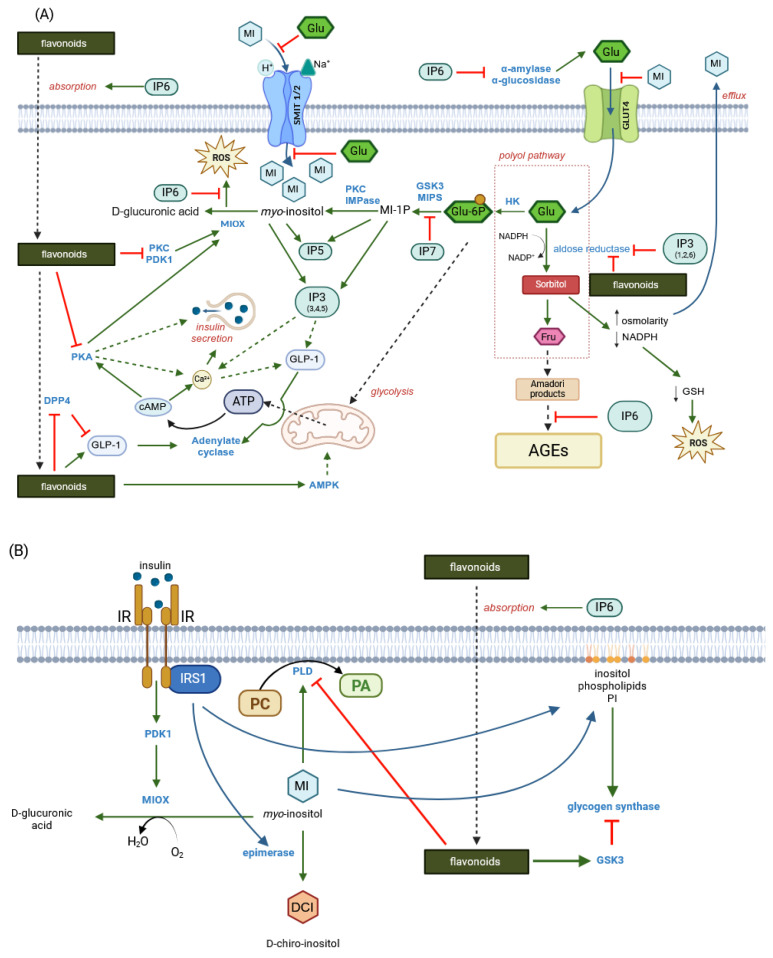
The impact of *myo*-inositol and *myo*-inositol phosphates on the intracellular signal transduction pathways. (**A**) the interactions between flavonoids and inositol phosphates in modulating enzyme activity; (**B**) the interplay between inositol phosphates and insulin signalling. AGEs—advanced glycation end products; AMPK—AMP-dependent protein kinase; DPP4—dipeptidyl peptidase-4; Fru—fructose; GLP-1─glucagon-like peptide-1; Glu—glucose; Glu-6P—glucose 6-phosphate; GSH—glutathione; GSK3—glycogen synthase kinase 3; HK—hexokinase; IMPase—inositol monophosphatase; IP3, IP5, IP6, IP7—inositol phosphates; IR—insulin receptor; IRS-1—insulin receptor substrate 1; MI—*myo*-inositol; MI-1P—*myo*-inositol-1-phosphate; MIOX—*myo*-inositol oxygenase; MIPS—*myo*-inositol 1-phosphate synthase; PA—phosphatidic acid; PC—phosphatidylcholine; PI—phosphatidylinositol; PDK1—phosphoinositide-dependent kinase 1; PKA—protein kinase A; PKC—protein kinase C; PLD—phospholipase D; ROS—reactive oxygen species; SMIT ½—sodium *myo*-inositol cotransporter-1/2 (based on [88,89,90,91,92,93,94,95,96,97]) (created in BioRender. Duda, A. (2025), https://BioRender.com/z92k205 accessed on 20 December 2024).

Based on data obtained from β-cells, models have been developed to explain oscillatory responses to GLP-1 generated by sequential positive and negative feedback regulation due to rapid activation and slow inhibition of the IP3 receptor (IP3R) by Ca^2+^ [98]. The question therefore arises as to whether inositol triphosphates generated in the gastrointestinal tract from foods high in both phytate and endogenous phytases [48] or by microbial phytases [99] could be an agonist of the IP3R. Results from studies in monogastric animals seem to support such a possibility. In the aforementioned paper by Schlemmer et al. [48], higher inositol phosphates (IP4-6) precipitated in the small intestine and reduced the availability of metal ions for absorption, while IP3 remained soluble. There was no further phytate hydrolysis by plant phytases in the duodenum and ileum. In the colon, further phytate hydrolysis was attributed to intestinal microflora, which may secrete both 3- and 6-phytases. Furthermore, the high concentration of IP6 in small intestinal chyme observed during the digestion of phytase-free diets was effectively hydrolysed in the colon. It, therefore, appears that, irrespective of the level of phytate degrading activities in foods or dietary ingredients, substantial IP6 degradation takes place in the intestinal tract of monogastric animals, but that those ingredients rich in endogenous phytases, such as most cereal grains, allow at least partial control of the dephosphorylation pathway so that IP3(1,2,3), with well-established antioxidant activity [100], accumulates in the small intestine. It was claimed [101] that these compounds must have been absorbed, as the presence of IP3(1,2,3) in mammalian cells at levels ranging from 0.6 to 13.1 μM. However, the most intriguing hypothesis has its roots in the findings of Wu et al. [99] who found that a signalling molecule IP3(1,4,5) is released from phytate by commensal bacteria, including *Escherichia coli*. Intestinal epithelial cells from mice exposed to microbiota showed induction of genes predicted to be transcribed in the presence of elevated IP3 levels, and phytate ingestion promoted recovery from intestinal damage. The authors provided a simple model of intestinal homeostasis and repair in which epithelial HDAC3 functions as a sensor for several microbiota-derived metabolites that are activated by inositol phosphates released from phytate. These substances appear to prevent the inhibition of intestinal HDAC3 activity by butyrate, a microbial metabolite of dietary fibre. In addition, IP3 released from phytate by commensal bacteria may promote the synthesis of GLP-1. In the intestine of rats with short bowel syndrome, phytate increased IP3 levels, which stimulated HDAC3 activity, but did not alter HDAC3 gene expression levels [102] (Figure 5). Increased HDAC3 activates the transcription factor STAT3 and the synthesis of antimicrobial proteins (AMPs). In this study, rice bran and its component phytate also increased intestinal length and mucosal weight, while IP3 was shown to stimulate intestinal epithelial cell proliferation and decrease intestinal permeability. These effects were inhibited by 10 mM butyrate. Epithelial STAT3 activation by rice bran or other phytate-rich foods has been shown to prime epithelial defence against infection and host immunity [25]. There is also evidence that dietary phytate increases the expression of another histone deacylase, SIRT-1, in APAP-induced hepatoxicity, either through iron chelation and attenuation of oxidative stress or through direct SIRT-1 up-regulation [103]. However, inositol tetraphosphate IP4(1,4,5,6), but not IP3(1,4,5), directly activated HDACs and, therefore, a wide range of nuclear events must have been under the control of inositol polyphosphate multikinase (IPMK) [104,105].

On the other hand, free dietary *myo*-inositol (MI) did not alter HDAC3 activity. A metabolic pathway allowing the conversion of phytate-derived MI to propionate and acetate by the commensal *Anaerostipes rhamnosivorans* has been described [106]. *A. rhamnosivorans* induced propionate CoA transferase and *myo*-inositol 2-dehydrogenase enzymes that converted both MI and D-chiro-inositol to propionate after adaptation to inositol-containing substrates. The ability to accumulate propionate was enhanced by the cross-feeding mechanism between *A. rhamnosivorans* and *Bifidobacterium longum*, which expressed phytase activity. This idea is supported by another study [107], which identified *Mitsuokella jalaludinii* as an efficient phytase producer that worked synergistically with *A. rhamnosivorans* in the synthesis of propionate (Figure 5). These intestinal bacteria not only anaerobically converted inositol to propionate, but also reduced fasting glucose, insulin and triglyceride levels in the blood of mice. Propionate, like butyrate, regulated hepatic gluconeogenesis and inhibited intestinal histone deacetylases [108]. Butyrate produced by the gut microbiota, acting as an HDAC3 inhibitor, has been shown to improve the innate response of macrophages and clearance of pathogenic bacteria without inducing tissue-destructive inflammation [109]. It, therefore, appears that the excess of MI, either due to increased consumption or catalytic action of endogenous cereal phytases, may provide additional health benefits in the presence of specialised microbiota.

To gain an even broader picture of how IP6 consumption affects the gut microbiome, the metabolism of a prominent gut bacterium, *Bacteroides thetaiotaomicron*, must also be considered. *B. thetaiotaomicron* expresses a bacterial homologue of mammalian IP6 phosphatase (MINPP), an enzyme with the high catalytic activity that generates IP3 and consequently stimulates intracellular Ca^2+^ release in human colonic epithelial cells [110]. The MINPP family of bacterial enzymes contributes to IP6 homeostasis in the mammalian gastrointestinal tract. HDAC3 activity and its epigenetic function play a crucial role in maintaining intestinal epithelial cell homeostasis, i.e., it controls Paneth cell development and the production of cytokines and antimicrobial peptides such as lysozyme. It also regulates mucosal interactions with commensal microbiota and modifies the composition of the gut microbial community [111,112]. Thus, microbiota may contribute to intestinal homeostasis not only by modulating HDAC3 activity through phytate-mediated metabolism [113], but also by participating in the interconversions between signalling inositol phosphates at the epithelial level.

Taken together, accumulating evidence suggests that inositol and inositol phosphates released by phytate contribute to the epigenetic control important for intestinal mucosal homeostasis by differentially affecting the activity of HDAC3. These mechanisms maintain a fragile balance in the transcriptional regulation of the gene patterns responsible for mucosal regeneration, differentiation and immunological protective functions (as summarised in Figure 5).

### 4.2. Intestinal Absorption Phenomena and Regulation of Tight Junction Transport—Two Mechanisms: Tight Junction Modulation and P-Glycoprotein Inhibition

Tight junctions are now recognised as multifunctional molecular gates responsible for epithelial barrier integrity and intestinal permeability that are modulated by the gut microbiota and implicated in many degenerative diseases resulting from systemic inflammation [114]. Although new methods including genome editing and new research approaches have been applied, the classical studies on these structures are based on the use of low Ca^2+^ treatments [115,116,117]. Fu et al. [22], using Caco-2 cell monolayers to study IP6 as an enhancer of flavonoid absorption, reported a reversible downregulation of the expression of tight junction proteins, namely claudin-1, occludin and ZO-1, possibly due to chelation of Ca^2+^ and Mg^2+^ ions (Figure 6). Furthermore, α-lactalbumin, a small milk protein with a strong Ca^2+^-binding site [118], has been found to modulate tight junction permeability, allowing enhanced uptake of *myo*-inositol [119]. It is, therefore, reasonable to speculate that phytic acid may exert an enhancing effect on the absorption of *myo*-inositol, a substance generated from phytate as a result of its complete dephosphorylation. The results of our studies in poultry seem to support this hypothesis, as the effects of phytate-released inositol were different from those of pure supplemental *myo*-inositol [120]. Notably, intestinal permeability may also be modulated by IP3(1,4,5), whose elevated levels result not only from activation of phospholipase C, but also from phytate degradation by the gut microbiota. Increased intracellular calcium and myosin light chain kinase would promote actin–myosin filament contraction and tight junction opening [121]. A fundamental issue in our understanding of the various modes of regulation of intestinal tight junctions by phytate relates to the fact that, in addition to reducing the expression of claudin-1, IP6 has been documented to stimulate the mRNA and protein synthesis of intestinal mucus (MUC2, trefoil factor 3; TFF3) and E-cadherin—the key adhesion junction proteins that maintain the structural integrity of the epithelial cell layer. In rats used as a model of colorectal cancer, IP6 also reduced serum levels of pro-inflammatory cytokines and generally ameliorated intestinal barrier damage [122].

The assembly and expression of tight junction proteins were also enhanced by quercetin [127], and the beneficial effects of curcumin, quercetin, resveratrol and epigallocatechin gallate on mucosal barrier integrity in inflammatory bowel disease have been documented in animal and clinical studies [128,129]. Furthermore, ferulic acid conjugated with octopamine ameliorated intestinal barrier disruption and inflammatory responses in inflammatory bowel disease [130]. Protective effects on the epithelial barrier and reduction in colonic inflammation have also been attributed to phenolic acids in wheat bran, including vanillic acid, ferulic acid, caffeic acid, gallic acid and protocatechuic acid [131]. Microbial fermentations of wheat bran by bacteria and yeasts significantly increased the concentration of free phenolic acids, while phytic acid was hydrolysed by endogenous wheat phytase, releasing IP3(1,2,3) with antioxidant functions [132]. Dietary fibre from rice and wheat bran stimulated the abundance of mucus-associated bacteria and SCFA-producing bacteria, improved the intestinal barrier and regulated the immune system by downregulating levels of inflammatory cytokines [133]. Strengthening of the intestinal barrier, together with a reduction in oxidative stress, has also been attributed to the bound polyphenols of oat bran, the constituents of which simultaneously altered the composition of the microbiota, which differed from the microbial populations associated with elevated levels of free polyphenols [134]. There appears to be a well-documented rationale for the idea that cereal phenolics and phytate have a beneficial effect on epithelial cell integrity, but in addition, phytate and some of its hydrolysis products regulate tight junction opening mechanisms controlled by changing Ca^2+^ concentrations.

P-glycoprotein (Pgp), a member of the ABC (ATP-binding cassette) transporter superfamily, can bind specifically to its substrates to pump them out of the cell. It is therefore necessary to co-administer a P-glycoprotein inhibitor to maintain a bioactive compound inside the cell [135]. Phytate was found to be a non-competitive P-gp inhibitor with the potential to enhance the uptake of bioactive compounds—substrates of P-glycoprotein. This was supported by the observation that the decrease in Pgp activity in the presence of IP6 was not only due to conformational changes induced by IP6, but also due to a significant decrease in Pgp mRNA expression [28]. Tian and co-workers [136] designed phytic acid-modified CeO_2_ nanorods for Ca^2+^ scavenging to apply the Ca^2+^-negative regulation strategy of Pgp. Flavonoids can be classified into Pgp substrates, whose efflux from the cell reduces their absorption, and Pgp inhibitors (including quercetin, rutin and tangeretin), which increase the bioavailability of drugs and nutrients [137]. In order to maximise the beneficial effects of these bioactives, it has been suggested that a mixture of specific flavonoids containing both Pgp substrates and Pgp inhibitors should be developed to stimulate their absorption. It has also been suggested that some flavonoids bind to the Pgp binding site, while others have the ability to interact with the ATP binding site of Pgp and exert an inhibitory effect. Of note, ferulic acid, the major phenolic acid of cereals, was shown to inhibit the PI3K/Akt/NF-κB signalling pathway and reverse Pgp-mediated multidrug resistance [138]. In mice fed a high-fat diet, ferulic acid increased the population of SCFA-producing bacteria, inhibited endotoxin generation, modulated tight junction protein expression, alleviated colonic barrier dysfunction and reduced intestinal inflammation [139]. It can therefore be postulated that epithelial barrier integrity and intestinal permeability, the key factors determining the bioavailability of cereal components, are modulated by both phenolics and phytate, with an overwhelming role of the gut microbiota.

## 5. Colonic Fermentations, Regulation, Breakdown of Polyphenols and Inositol Phosphates

There seems to be a general consensus that digesta passes from the small intestine to the colon where microbiota ferment undigested carbohydrates and possibly other nutrients, producing SCFAs, secondary bile acids, neurotransmitters such as GABA, dopamine, serotonin, glutamate, B vitamins and lowering colonic pH [140]. A further consensus in the field is that SCFAs exert multiple beneficial effects, in particular by modulating immune function, thereby ameliorating host susceptibility to various diseases [141,142]. In the colonic epithelium, SCFAs mediated by specific transporters affect the integrity and permeability of the intestinal barrier mainly by upregulating genes encoding tight junction and mucus layer proteins (Figure 5). One of the best-established functions of butyrate is the modulation of oxidative stress [143]. A fundamental question in our understanding of the role of cereal constituents in exerting health-promoting activities is whether and how phenolics, particularly ferulic acid and certain inositol phosphates, are involved in the regulation of oxidative stress. There is strong evidence that both ferulic acid and phytic acid exert antioxidant activity, but, in addition, both cereal components have been shown to have anti-inflammatory, antidegenerative and anticancer functions [144,145]. Although many parallels can be drawn between the mechanisms governing the health-promoting effects of phytic acid and ferulic acid, e.g., both have been shown to suppress colon cancer [146,147], ferulic acid is best known for its potent direct free radical scavenging activity and up-regulation of antioxidant enzymes [145]. Phytic acid, while preserving the solubility of iron, almost completely blocks the superoxide-driven generation of ^•^OH radicals [148]. Furthermore, in addition to phytic acid, iron chelates of IP3(1,2,3) and IP4(1,2,3,x), which occupy all six coordination sites on iron, inhibited oxidation and peroxidation reactions due to a unique 1,2,3 (equatorial–axial–equatorial) conformational motif. *Myo*-inositol phosphates with the conserved IP(1,2,3…) motif, released from phytate by the action of phytases, could therefore be considered as an antioxidant family of IPx that would maximise the beneficial effects of cereal consumption (Figure 7).

In the course of phytate hydrolysis, endogenous 6-phytases of rye, barley, spelt, oats, wheat and rice were found to generate the antioxidant IP5, IP4 and IP3 [152].

Among the colonic microbiota, certain bifidobacteria, particularly *Bifidobacterium longum* spp. *infantis* and *Bifidobacterium pseudocatenulatum*, increased levels of the antioxidant inositol triphosphate [153] and increased zinc solubility [154]. A nutraceutical formulation of wheat bran and millet (1:1) was found to induce the biosynthesis of ferulic acid esterase, xylanase and pectinase enzymes by the gut microbiota during in vitro fermentation and to significantly increase the relative abundance of *Bifidobacterium* in the fermentation medium [155]. At this point in the review of current knowledge on cereal bioactivities, at least two conclusions can be drawn: Colonic microbiota has the potential to biosynthesise enzymatic activities (pectinase, xylanase, phytase, ferulic esterase) similar to endogenous cereal enzymes and to improve nutrient accessibility at the final stage of digestion. Secondly, microbial phytases and ferulic esterases active in the large intestine hydrolyse nutrients to produce compounds with exceptional antioxidant activity (Figure 7). Endogenous cereal enzymes appear to exert similar effects at earlier stages of digestion. It should be emphasised that the strong antioxidant activity of cereal phenolics and inositol phosphates implies the ability of these compounds to inhibit the formation of advanced glycation end products (AGE) (Figure 4A). Finally, fibre-rich diets in general, and whole grain and bran-rich diets in particular, have been proposed as simple tools for microbiome modification to increase the abundance of *Lactobacillus* spp. and *Bifidobacterium* spp. in the colon [156]. It should be emphasised that the aforementioned microbiome metabolites not only affect immune function, i.e., by modulating inflammation, but are also important in the stress response, mood, emotion, wound healing and appetite regulation.

## 6. *Myo*-Inositol

It is well known that phytate dephosphorylation by cereal or microbial phytases proceeds in a stepwise fashion, yielding mixtures of non-hydrolysed IP6, inositol phosphates (IPx) and free *myo*-inositol (MI) (Figure 6). In cereals, the unique composition, which can include both antioxidant and/or signalling IPx, serves as a scaffold for a multifunctional network of bioactive substances including IP6, IPx, MI and phenolic compounds. In particular, the bioactivity of these compounds may be manifested through the inhibition of aldose reductase (AR), a key enzyme in the polyol pathway that is activated in a state of hyperglycaemia. Inositol triphosphate IP3(1,2,6)—one of the products of cereal and microbial phytases—has been found to be a potent inhibitor of AR [157] and the same function has been attributed to ferulic acid [158,159]. Aldose reductase-mediated polyol accumulation in various tissues contributes to a number of human diseases and is associated with oxidative stress, DNA damage, AGE formation and MI depletion [160]. Multiple mechanisms regulate MI homeostasis and high glucose diets, in particular, reduce MI bioavailability by inhibiting its biosynthesis and absorption, and stimulate its degradation by *myo*-inositol oxygenase (MIOX) [161]. *Myo*-inositol depletion has been implicated as a major cause of cataractogenesis [162], bipolar disorder [163] and other metabolic deregulations common in Western societies [164].

One of the fundamental questions in our understanding of cereal bioactivities is whether MI generated from foods rich in IP6, endogenous phytases and natural antioxidants such as phenolic acids, tannins, coumarins and terpenoid derivatives [162] can not only provide bioaccessible inhibitors of AR and supply exogenous dietary MI, but also provide soluble IP6—a potent inhibitor of ROS and AGEs [165]—which also inhibits intestinal digestive enzymes such as α-amylase, α-glucosidase and pancreatic lipase, as discussed above. A concept of MI released from phytate by the action of phytase and accompanied by various IPx and also by non-hydrolysed IP6, the so-called enzymatically generated inositol (EGI; Figure 8), has been exploited in poultry diets [166]. Our studies showed that MI released from dietary phytate can exert specific effects in different tissues and, in particular, inositol generated by different phytases in the gut modulated avian physiology in a different way than pure, supplemental MI added to the feed. However, a large body of evidence suggests that independent of exogenous inositol availability, de novo synthesis of *myo*-inositol occurs in mammalian cells and, thus, independent metabolic pathways provide MI for two distinct functions, i.e., osmoregulation and synthesis of phosphorylated signalling compounds [161]. DiNicolantonio and O’Keefe [164], reviewing the role of *myo*-inositol in insulin signalling and glucose metabolism, noted that cellular MI can be recycled from IP3 and IP2 and, therefore, increased dietary intake of MI is likely to ‘conserve’ IP3 and IP2. Similar theses regarding the physiological role of the inositol phosphate signalling network have been proposed in another study [167]. It, therefore, appears that although antioxidant and signalling IP3s may be effective in the gut, IPx is dephosphorylated to MI prior to absorption into the systemic circulation. The idea that tissue IPx does not result from direct absorption is also supported by the findings of Sprigg and co-workers [168], who found different isomers of IP4 and IP5 in broiler liver and kidney from those known to be generated by *E. coli* 6-phytase added to the diet at a super-dose of 6000 FTU/kg.

Finally, it should be emphasised that both *myo*-inositol and ferulic acid belong to the group of cereal constituents for which there is strong evidence that, as dietary supplements, they effectively suppress high glucose levels and alleviate insulin resistance in various dietary models that mimic the action of metformin. *Myo*-inositol, like metformin, improved hyperglycaemia and dyslipidaemia in diabetic mice [169] and significantly improved glycaemic response and insulin resistance in obese patients with non-alcoholic fatty liver syndrome by acting on the AMPK/PI3K/AKT pathway [170]. For its part, ferulic acid showed antidiabetic properties by modulating insulin-signalling molecules in high-fat and fructose-induced type 2 diabetic rats [171] and maintained pancreatic ultrastructural architecture, improving glucose homeostasis in similar models [172]. In the aforementioned studies, experimental supplement concentrations ranged from 50 mg of ferulic acid per kg body weight in experimental rats to 2 g MI per day in patients with NAFLD. Such levels may have been considered unrealistically high compared to the concentrations of these bioactives in cereal grains [173], normally perceived as foods high in *myo*-inositol [174], but there is strong evidence that daily consumption of 100 g [175] to 150 g [176] of whole grains provides a wide range of protective benefits, including against type 2 diabetes. However, in order to provide a more accurate review of the current knowledge of the mechanisms governing intestinal homeostasis, the ability of the gut microbiota to degrade MI must be considered [177]. Therefore, it seems that in addition to a sufficient daily intake of cereal bioactives, appropriate tools to modulate the gut microbiota are necessary to maximise their beneficial effects.

Although several works have proposed strategies to reduce the phytate content in cereals in order to improve the bioaccessibility of minerals [178], our proposals are the opposite. Here, we propose strategies that would include work to optimise the functional properties of cereal foods that would ensure increased levels of IP6, IPx, MI and phenolic acids, together with enhanced endogenous activities of phytases and feruloyl esterase in cereal grains. To this end, well-established management strategies such as soaking, germination or fermentation should be prioritised, as suggested by Li et al. [179], but, unlike the authors of this comprehensive review, we would envisage up-regulation of phytic acid and phenolic biosynthesis in cereals. Furthermore, another key point in the strategy to maximise the beneficial effects of cereals in human nutrition is the selection of cereals with increased levels of genes encoding ferulic acid esterase, phytase and endoxylanase (Figure 9). In parallel, knowledge of the modulation of the gut microbiota by cereal components should be accumulated and tools to influence the composition and metabolism of the gut microbiota should be better established.

## 7. Conclusions

Current knowledge provides strong evidence for many possible synergistic effects between phytic acid, *myo*-inositol phosphates, free *myo*-inositol and ferulic acid. For example, both IP6 and FA exert potent and complementary antioxidant effects in the intestinal tract, maintaining cell barrier integrity and epithelial antibacterial immunity. IP6 enhances the absorption of nutrients, including phenolics, by regulating tight junctions and may act as a non-competitive P-glycoprotein inhibitor, while ferulic acid has been shown to reverse P-glycoprotein-mediated resistance and increase nutrient bioavailability. Ferulic acid, together with various *myo*-inositol phosphates resulting from increased hydrolysis of IP6 by endogenous and microbial phytases, would affect pepsin and gastric lipase activity and then incretin secretion, affecting insulin and glucagon release. Ferulic acid was found to bind to the GLP-1 molecule in a 1:1 ratio, effectively preventing hydrolysis of the hormone by DPP4, while IP3, a product of IP6 hydrolysis, may promote GLP-1 synthesis. Inositol phosphates, especially IP3(1,2,6), together with free *myo*-inositol generated by prolonged hydrolysis of phytate, would inhibit the aldose reductase enzyme and thus prevent various diabetic complications.The bran fraction of wheat, maize, brown rice and other cereals contains not only high levels of phytate, free and total phenolics, but also endogenous enzymes such as amylases, phytase, xylanase, β-glucanase and feruloyl esterase, whose activities can be increased by germination. In addition, cereals in the form of wholemeal flour or bran can be subjected to enzymolysis by exogenous enzyme preparations to achieve advanced hydrolysis of cereal phytate and phenolic compounds, further enhanced by biotransformations from gut microbiota.Here, we have proposed a novel strategy of selecting cereals with high phytate, phenolic and endogenous phytase, ferulic esterase and endoxylanase activities as a starting point for strategies to produce value-added health-promoting foods. The key assumption of the strategy is the advanced hydrolysis of phytate and phenolic compounds by cereal and/or microbial enzymes, which would generate substantial amounts of “enzymatically generated inositol” (EGI), which includes phytic acid, myo-inositol phosphates and myo-inositol, the compounds that, together with free ferulic acid, provide enhanced bioavailability of cereal nutrients through multiple synergistic effects.

## Figures and Tables

**Figure 1 molecules-30-00652-f001:**
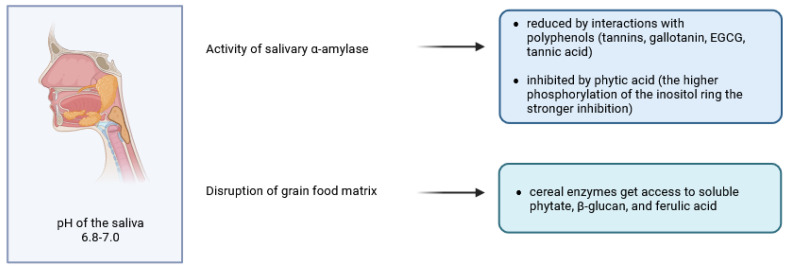
Oral cavity starting point of cereal digestion reactions involving phytates and polyphenols (created in BioRender. Duda, A. (2025), https://BioRender.com/z92k205 (accessed on 20 December 2024)).

**Figure 2 molecules-30-00652-f002:**
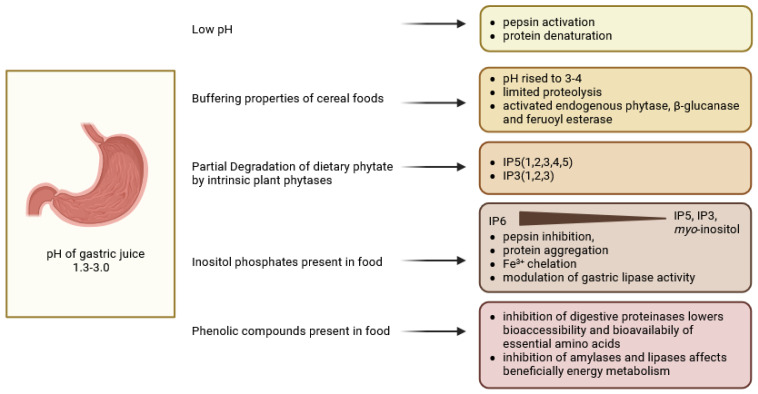
Stomach digestion effects on phytates and polyphenols (created in BioRender. Duda, A. (2025), https://BioRender.com/z92k205 (accessed on 20 December 2024)).

**Figure 3 molecules-30-00652-f003:**
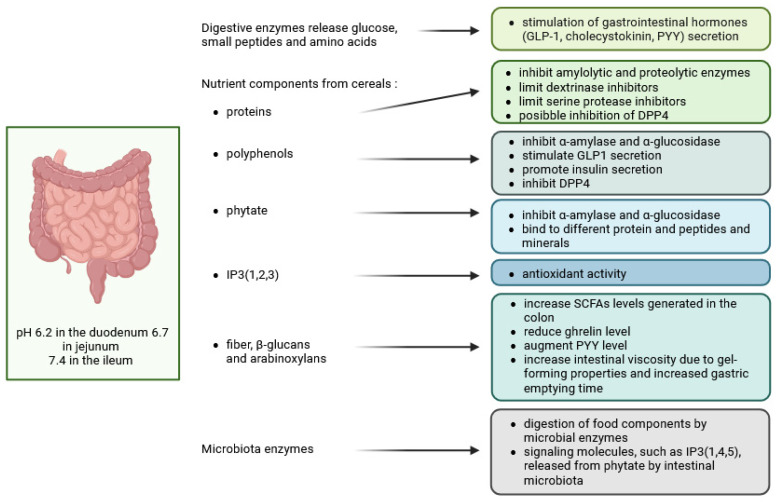
Mechanisms affecting bioaccessibility and bioavailability of cereal nutrients in intestinal digestion (created in BioRender. Duda, A. (2025), https://BioRender.com/z92k205 accessed on 20 December 2024).

**Figure 5 molecules-30-00652-f005:**
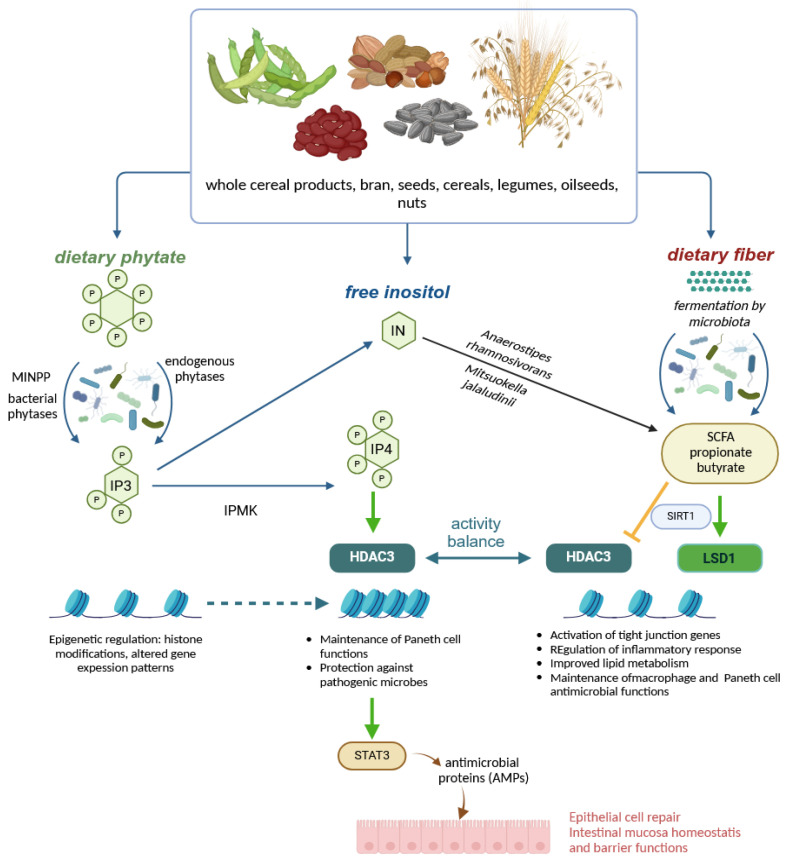
The food sources of phytate and inositol phosphates and their impact on the intestinal mucosa homeostasis, including the epigenetic regulation of gene expression. HDAC3—histone deacetylase 3; IN—free inositol; IP—inositol phosphates; LSD1—lysine-specific demethylase 1; MINPP—mammalian IP6 phosphatase; SCFA—short-chain fatty acids (created in BioRender. Duda, A. (2025), https://BioRender.com/z92k205 accessed on 20 December 2024).

**Figure 6 molecules-30-00652-f006:**
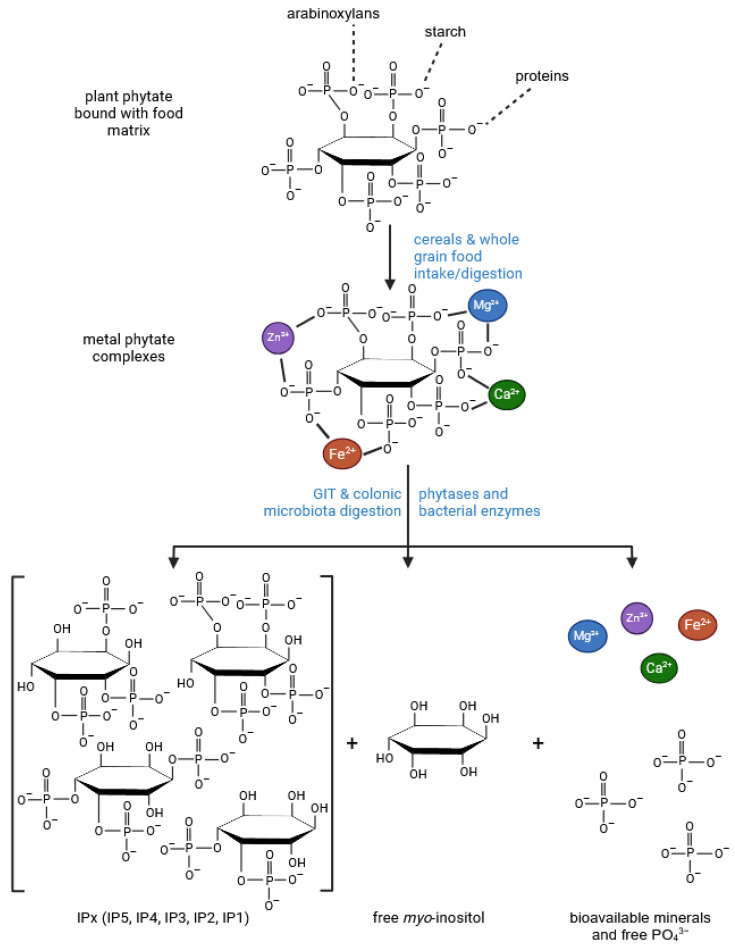
Release of phytic acid from the food matrix, hydrolysis of phytate metal complexes and biosynthesis of inositol phosphates and free *myo*-inositol by endogenous and microbial phytases (based on [123,124,125,126]) (created in BioRender, Duda, A. (2025), https://BioRender.com/z92k205 accessed on 20 December 2024).

**Figure 7 molecules-30-00652-f007:**
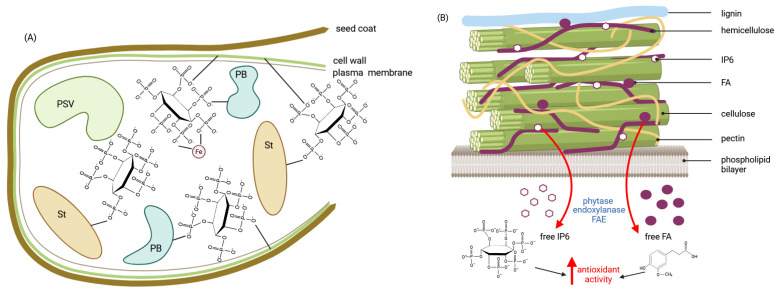
The release of phytic acid and free polyphenols from the food matrix (cereals); (**A**) barley seed with phytate and ferulic acid bond to the cell wall components, starch and protein; (**B**) zoom view of the cereal cell wall structure. FA—ferulic acid; IP6—inositol hexakisphosphate; PSV—protein storage vacuole; St—starch; PB—protein body (based on [149,150,151]) (created in BioRender. Duda, A. (2025), https://BioRender.com/z92k205 accessed on 20 December 2024).

**Figure 8 molecules-30-00652-f008:**
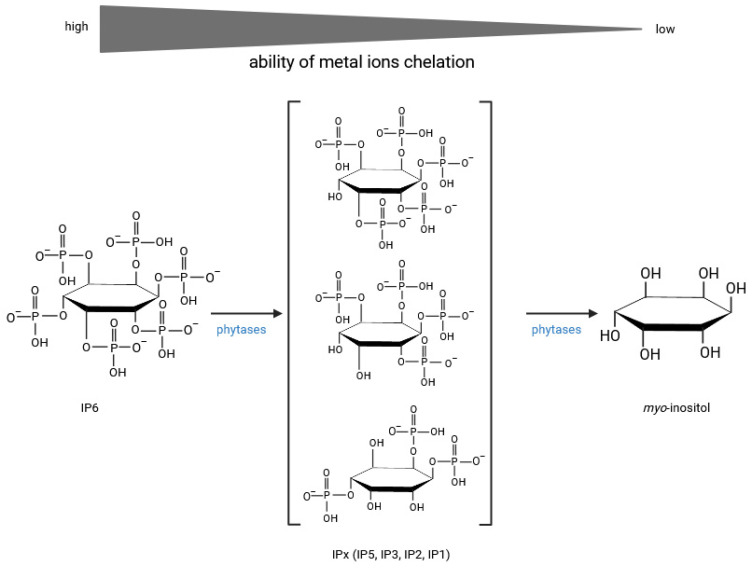
The concept of enzymatically generated inositol (EGI) that comprises a mixture of unhydrolysed IP6, IPx and *myo*-inositol (created in BioRender. Duda, A. (2025), https://BioRender.com/z92k205 accessed on 20 December 2024).

**Figure 9 molecules-30-00652-f009:**
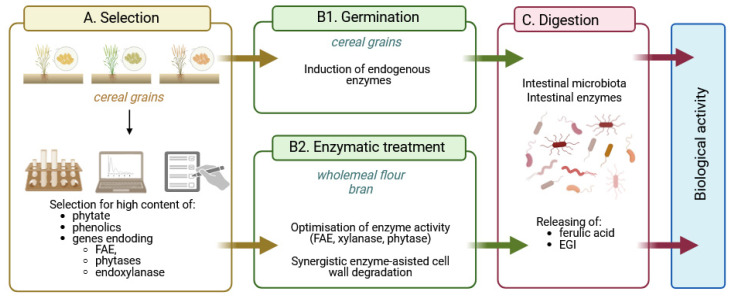
Strategies suggestion for optimising the functional properties of cereal foods with increased amounts of EGI and phenolic acids exerting different health-promoting bioactivities (created in BioRender. Duda, A. (2025), https://BioRender.com/z92k205 accessed on 20 December 2024).

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
