# Peer review of "Towards Improved Bioavailability of Cereal Inositol Phosphates, Myo-Inositol and Phenolic Acids"

_molecules, 2025, doi:10.3390/molecules30030652_

Round 1

Reviewer 1 Report

Comments and Suggestions for Authors

The review is rather comprehensive and coherent, and the figures greatly aid in illustrating and clarifying the presented concept. I would suggest publication after minor corrections, detailed below.

Line - comment

13. In general, the myo in myo-inositol should be in italics (see IUPAC reccomendations)

28. in THIS review

Section 2.

Correct me if I am wrong, but all referenced studies are in vitro. the way the paragraph is written, implies that before line 134, results are in vitro, while after are not. This gives an impresison of more certtainty that can be defined with in vitro test. Please do not conclude in vivo activities from in vitro data

102, in vitro needs italics

130, specificy the interactions

147: are the residence times enough to achive the degradation?

Line 170, should be numbered 3

174-176 needs a reference

177. You cannot have your cake and eat it too. please be more clear on whether the cereals rise or not the pH, because then that would follow which enzymes can be active. it must also be remembered that enzymes do work outside their optimal pH, which is part of the complexity of studying these biological systems

201- THE secondary...

217 elderlY...

228 In vitro in italics

234 same, from this point on, please double check use of italics

415: the D should be in small capitals format.

FIGURES:

This is perhaps an editorial issue, but for example, in Figure 6, 7 and 8 the phytate does not present a good resolutions. just make sure the figures are legible in the final PDF.

Author Response

Comments 1: The review is rather comprehensive and coherent, and the figures greatly aid in illustrating and clarifying the presented concept. I would suggest publication after minor corrections, detailed below.

Response 1: The Authors express their gratitude to the Reviewer I, both for high evaluation and the corrections made to improve the quality of the manuscript. We have introduced suggested corrections and all changes made in the manuscript are visible (revised text is in the mode ‘Track changes’).

Line - comment

Comments 2: 13. In general, the myo in myo-inositol should be in italics (see IUPAC reccomendations)

Response 2: This has been corrected in text and figures.

Comments 3:28. in THIS review

Response 3: Correction has been made. 

Comments 4: Section 2. Correct me if I am wrong, but all referenced studies are in vitro. the way the paragraph is written, implies that before line 134, results are in vitro, while after are not. This gives an impresison of more certtainty that can be defined with in vitro test. Please do not conclude in vivo activities from in vitro data

Response 4: Appropriate modifications have been introduced.

Comments 5: 102, in vitro needs italics

Response 5: Correction has been made.

Comments 6: 130, specificy the interactions

Response 6: The interactions have been specified.

Comments 7: 147: are the residence times enough to achive the degradation?

Response 7: Most probably the time is sufficient to, et least, start the degradation. Correction has been introduced.

Comments 8: Line 170, should be numbered 3

Response 8: Correction has been made.

Comments 9: 174-176 needs a reference

Response 9: Reference 46 has been introduced.

Comments 10: 177. You cannot have your cake and eat it too. please be more clear on whether the cereals rise or not the pH, because then that would follow which enzymes can be active. it must also be remembered that enzymes do work outside their optimal pH, which is part of the complexity of studying these biological systems

Response 10: We agree, appropriate amendments have been made within the text.

Comments 11: 201- THE secondary...

Response 11: Correction has been made.

Comments 12: 217 elderlY...

Response 12: Correction has been made.

Comments 13: 228 In vitro in italics

Response 13: Correction has been made.

Comments 14: 234 same, from this point on, please double check use of italics

Response 14: Corrections have been made.

Comments 15: 415: the D should be in small capitals format.

Response 15: Correction has been made.

Comments 16: FIGURES: This is perhaps an editorial issue, but for example, in Figure 6, 7 and 8 the phytate does not present a good resolutions. just make sure the figures are legible in the final PDF.

Response 16: The figures have been revised and improved to make them more readable. 

Reviewer 2 Report

Comments and Suggestions for Authors

The paper discusses cereals inositol phosphates, myo-inositol and phenolic acids and their roles in human health.

Unfortunately, in this form the document has numerous drawbacks and needs a major revision, hence please consider the following issues to address:

-        The abstract lacks clarity and is somewhat verbose; it should be more concise, focusing on key issues and implications. It should briefly mention why this work is relevant and how it contributes to existing knowledge.

-        The introduction fails to clearly articulate the specific research gap that the study aims to address, which is important for justifying the study's relevance and importance in advancing the field. The objectives of the study are not explicitly stated, making it difficult to understand the focus of the research. The flow of ideas in the introduction is somewhat disjointed, which can disrupt the reader's understanding; improving the logical progression of concepts and ensuring smooth transitions between paragraphs would enhance readability. Besides, there is a need to rephrase an important amount of this review, avoiding author names (filler style) – for instance “Abdulwaliyu et 91 al. [14], Vucenik [15], Brehm and Windhorst [16] reviewed the anticancer properties of  pure phytate, pointing to unique new options for the treatment of different types of cancer” should be “previous studies reviewed the anticancer properties of  pure phytate, pointing to unique new options for the treatment of different types of cancer [14-16] ”

-        The title of section 2 is too long – rephrase it. The connection between the breakdown of phytates and polyphenols and the broader implications for health or nutrition is not clearly articulated; emphasizing the significance of these processes in the context of the study's objectives would enhance relevance.

-        Figure 1 – replace “alivary” > salivary or better remove this figures, since it is too simple and redundant with the text above – filler.

-        Section “2. Digestion in the stomach” should be #3.  The relationship between stomach digestion and subsequent digestive processes in the intestines is not clearly articulated; establishing a connection between stomach digestion and its impact on later stages of digestion would provide a more comprehensive understanding.

-        L.198-202 –delete this phrase together with reference [50] , since it is not related with cereals (filler).

-        Section “3. Digestion in the small intestine” should be #4. It lacks a discussion on the implications of small intestine digestion for overall health and nutrition; highlighting how these processes affects nutrient bioavailability and metabolic health would add relevance to the content.

-        Section “4. Colonic fermentations, regulation, breakdown of polyphenols and inositol phosphates” should be #5. The implications of colonic fermentation for gut health and overall well-being can be deduced but are not thoroughly explored; a more detailed discussion on how these processes influence immune function, inflammation, and metabolic health would enhance the relevance of the content. The flow of information is somewhat disjointed, which can hinder reader comprehension.

-        L.531 – delete “γ-aminobutyric acid” – acronym defined in l.226

-        Section “5. Myo-inositol” should be #6; while the section mentions various functions of myo-inositol, it does not adequately detail the mechanisms by which it exerts its effects, particularly in relation to insulin signaling and cellular signaling pathways. The section could benefit from a discussion on dietary sources of myo-inositol and factors affecting its bioavailability.

-        “Summary” is a useless section, since the paper starts with a summary called “abstract”. Instead, a concluding section is required, effectively encapsulating the main findings of this study, streamlining the content to focus on key takeaways without excessive detail for enhanced clarity and impact. The concluding section can benefit from a discussion on the broader implications of the findings and potential future research directions; highlighting the significance of the research in the context of health and nutrition would add depth to the conclusion and including recommendations based on the findings would enhance the study’s relevance and encourage further exploration of the discussed topics.

-        L.711-712 – useless acknowledgement since the mentioned person is co-author; eventually add a new position in “author contribution” – delete it

Author Response

Comments 1: The paper discusses cereals inositol phosphates, myo-inositol and phenolic acids and their roles in human health.

Unfortunately, in this form the document has numerous drawbacks and needs a major revision, hence please consider the following issues to address:

Response 1: Authors thank the Reviewer II for the time spent and efforts made to enhance the quality of the manuscript. All changes made in the manuscript are the mode ‘Track changes’. 

Comments 2: The abstract lacks clarity and is somewhat verbose; it should be more concise, focusing on key issues and implications. It should briefly mention why this work is relevant and how it contributes to existing knowledge.

Response 2: The abstract section has been re-written in accordance with the suggestions made by all of the reviewers. Also the rest of the manuscript (Conclusions section) were improved according to the suggestions of all the reviewers.

Comments 3: The introduction fails to clearly articulate the specific research gap that the study aims to address, which is important for justifying the study's relevance and importance in advancing the field. The objectives of the study are not explicitly stated, making it difficult to understand the focus of the research. The flow of ideas in the introduction is somewhat disjointed, which can disrupt the reader's understanding; improving the logical progression of concepts and ensuring smooth transitions between paragraphs would enhance readability.

Response 3: Two additional general remarks have been inserted into the Introduction section (or other parts of manuscript) to fill the gaps identified by the reviewers. 

Comments 4: Besides, there is a need to rephrase an important amount of this review, avoiding author names (filler style) – for instance “Abdulwaliyu et 91 al. [14], Vucenik [15], Brehm and Windhorst [16] reviewed the anticancer properties of  pure phytate, pointing to unique new options for the treatment of different types of cancer” should be “previous studies reviewed the anticancer properties of  pure phytate, pointing to unique new options for the treatment of different types of cancer [14-16] ”

Response 4: The manuscript has been corrected according this suggestion.

Comments 5: The title of section 2 is too long – rephrase it. The connection between the breakdown of phytates and polyphenols and the broader implications for health or nutrition is not clearly articulated; emphasizing the significance of these processes in the context of the study's objectives would enhance relevance.

Response 5: The title of the section 2 has been rephrased to: “The breakdown of cereal phytates and polyphenols in oral cavity”. We also try to emphasize in the whole manuscript the significance of these breakdown reactions.

Comments 6: Figure 1 – replace “alivary” > salivary or better remove this figures, since it is too simple and redundant with the text above – filler.

Response 6: Appropriate corrections have been made in figures, according to the suggestions made by all of the reviewers.

Comments 7: Section “2. Digestion in the stomach” should be #3.  The relationship between stomach digestion and subsequent digestive processes in the intestines is not clearly articulated; establishing a connection between stomach digestion and its impact on later stages of digestion would provide a more comprehensive understanding.

Response 7: The section number has been corrected. A phrase has been introduced to emphasise this relationship.

Comments 8: L.198-202 – delete this phrase together with reference [50] , since it is not related with cereals (filler).

Response 8: We do not share the view of the Reviewer as naringenin was also found in cereals (wheat, emmer wheat, barley). In order to emphasise this, an additional reference [51] has been introduced.

Comments 9: Section “3. Digestion in the small intestine” should be #4. It lacks a discussion on the implications of small intestine digestion for overall health and nutrition; highlighting how these processes affects nutrient bioavailability and metabolic health would add relevance to the content.

Response 9: The section number has been corrected. A sentence about gut-brain and gut-liver axes has been introduced.

Comments 10: Section “4. Colonic fermentations, regulation, breakdown of polyphenols and inositol phosphates” should be #5. The implications of colonic fermentation for gut health and overall well-being can be deduced but are not thoroughly explored; a more detailed discussion on how these processes influence immune function, inflammation, and metabolic health would enhance the relevance of the content. The flow of information is somewhat disjointed, which can hinder reader comprehension.

Response 10: The section number has been corrected. An additional sentence has been inserted.

Comments 11: L.531 – delete “γ-aminobutyric acid” – acronym defined in l.226

Response 11: The correction has been made.

Comments 12: Section “5. Myo-inositol” should be #6; while the section mentions various functions of myo-inositol, it does not adequately detail the mechanisms by which it exerts its effects, particularly in relation to insulin signaling and cellular signaling pathways. The section could benefit from a discussion on dietary sources of myo-inositol and factors affecting its bioavailability.

Response 12: The section number has been corrected. We have highlighted existing and added a new reference [163] to direct readers to existing reviews on insulin and cellular signalling by myo-inositol and myo-inositol phosphates. Similarly, a new reference has been added [171] to position cereals among foods high in myo-inositol.

Comments 13:  “Summary” is a useless section, since the paper starts with a summary called “abstract”. Instead, a concluding section is required, effectively encapsulating the main findings of this study, streamlining the content to focus on key takeaways without excessive detail for enhanced clarity and impact. The concluding section can benefit from a discussion on the broader implications of the findings and potential future research directions; highlighting the significance of the research in the context of health and nutrition would add depth to the conclusion and including recommendations based on the findings would enhance the study’s relevance and encourage further exploration of the discussed topics.

Response 13: The 'Summary' section has been replaced by a 'Conclusions' section in which we try to briefly highlight the most important elements of this review paper.

Comments 14: L.711-712 – useless acknowledgement since the mentioned person is co-author; eventually add a new position in “author contribution” – delete it

Response 14: The correction has been made.

Reviewer 3 Report

Comments and Suggestions for Authors

The manuscript titled "Towards improved bioavailability of cereal inositol phos-phates, myo-inositol and phenolic acids", provides a comprehensive review of the role of cereal inositol phosphates, myo-inositol, and phenolic acids in enhancing the bioavailability of cereal nutrients. Here are my comments and questions regarding the manuscript:

1.    The manuscript contains numerous errors in expression. Please carefully review and make corrections.

2.    I think the abstract section could be further condensed. And what specific gaps in knowledge regarding the bioavailability of cereal inositol phosphates, myo-inositol, and phenolic acids does this review aim to address?

3.    How do the activities of endogenous cereal enzymes like phytase, β-glucanase, and feruloyl esterase change during the oral and gastric phases of digestion, and what factors influence their activation and efficacy? Please answer and highlight in your manuscript.

4.    The review mentions the role of incretin hormones like GLP-1 in regulating insulin and glucagon release. How do inositol phosphates and phenolic acids influence the secretion of these hormones, and what are the underlying mechanisms involved?

5.    What is the significance of the interaction between inositol phosphates and P-glycoprotein in the context of nutrient absorption and drug bioavailability? Are there any potential implications for the co-administration of cereals with medications? Please answer and highlight in your manuscript.

Author Response

Comments 1: The manuscript titled "Towards improved bioavailability of cereal inositol phos-phates, myo-inositol and phenolic acids", provides a comprehensive review of the role of cereal inositol phosphates, myo-inositol, and phenolic acids in enhancing the bioavailability of cereal nutrients. Here are my comments and questions regarding the manuscript:

Response 1: The authors are grateful to the Reviewer for his/her thoughtful efforts to improve the quality of the manuscript. We have carefully reviewed the manuscript and corrected it according to all the Reviewers' suggestions. All changes made in the manuscript are in 'Track changes' mode.  

Comments 2.    The manuscript contains numerous errors in expression. Please carefully review and make corrections.

Response 2: We carefully check the manuscript and we hope there are no more such errors.

Comments 3.    I think the abstract section could be further condensed. And what specific gaps in knowledge regarding the bioavailability of cereal inositol phosphates, myo-inositol, and phenolic acids does this review aim to address?

Response 3: The abstract section has been re-written in accordance with the suggestions made by all of the reviewers.

Comments 4.    How do the activities of endogenous cereal enzymes like phytase, β-glucanase, and feruloyl esterase change during the oral and gastric phases of digestion, and what factors influence their activation and efficacy? Please answer and highlight in your manuscript.

Response 4: As we stated in the manuscript, endogenous cereal phytase, β-glucanase and feruloyl esterase require access to their respective substrates, sufficient water activity and a pH suitable for their catalytic action. If these conditions are met, Michalis-Menten kinetic relationships would apply. The first condition, however, is the release of endogenous enzyme molecules from their association with other tissue components. In the oral phase of digestion, the release of these enzymes and their substrates is at least partially achieved, and the components of saliva, particularly water, would certainly activate their catalytic action. However, due to the unfavorable pH (too high), the efficiency of the reactions will be low. During the first stage of digestion in the stomach, the digesta is acidified, which can bring the pH close to the optimum value (4-5). This, together with the advanced physical disintegration of the tissues, makes the gastric digestion phase the most important for the catalytic action of these enzymes. Most of these relationships have already been described in the original manuscript. However, we add a few complementary statements.          

Comments 5.    The review mentions the role of incretin hormones like GLP-1 in regulating insulin and glucagon release. How do inositol phosphates and phenolic acids influence the secretion of these hormones, and what are the underlying mechanisms involved?

Response 5: The modified Figure 4A shows a fragment illustrating the influence of polyphenols on GLP-1 through DPP4 modulation and the effects of IP3. In the body of the manuscript, we have highlighted the possibility that IP3 released from phytate by commensal bacteria may promote the synthesis of GLP-1.

Comments 6.    What is the significance of the interaction between inositol phosphates and P-glycoprotein in the context of nutrient absorption and drug bioavailability? Are there any potential implications for the co-administration of cereals with medications? Please answer and highlight in your manuscript.

Response 6: While pure inositol hexakisphosphate (phytic acid) is the well-established non-competitive inhibitor of P-glycoprotein, and as a tight junction modulator is perceived as an enhancer of nutrient absorption, which most likely also affects drug absorption, the cereal food matrix as a whole seems to be a more complex system, where co-administration with drugs may result in diverse effects. These effects would mainly depend on the methods of food processing and we believe that the strategy proposed in this manuscript may provide fully controllable beneficial food-drug interactions. However, intensive research is needed to support this premise.    

Round 2

Reviewer 3 Report

Comments and Suggestions for Authors

accepted